DICOM for quantitative imaging biomarker development: a standards based approach to sharing clinical data and structured PET/CT analysis results in head and neck cancer research

Fedorov Andriy 1 2 andrey.fedorov@gmail.com fedorov@bwh.harvard.edu
Clunie David 3
Ulrich Ethan 4 5
Bauer Christian 4 5
Wahle Andreas 4 5
Brown Bartley 6
Onken Michael 7
Riesmeier Jörg 8
Pieper Steve 9
Kikinis Ron 1 2 10 11
Buatti John 12
Beichel Reinhard R. 4 5 13
1 Department of Radiology, Brigham and Women’s Hospital , Boston, MA , United States of America
2 Harvard Medical School, Harvard University , Boston, MA , United States of America
3 PixelMed Publishing, LLC , Bangor, PA , United States of America
4 Department of Electrical and Computer Engineering, University of Iowa , Iowa City, IA , United States of America
5 Iowa Institute for Biomedical Imaging, University of Iowa , Iowa City, IA , United States of America
6 Center for Bioinformatics and Computational Biology, University of Iowa , Iowa City, IA , United States of America
7 OpenConnections GmbH , Oldenburg , Germany
8 Freelancer , Oldenburg , Germany
9 Isomics, Inc. , Cambridge, MA , United States of America
10 Fraunhofer MEVIS , Bremen , Germany
11 Mathematics/Computer Science Faculty, University of Bremen , Bremen , Germany
12 Department of Radiation Oncology, University of Iowa Carver College of Medicine , Iowa City, IA , United States of America
13 Department of Internal Medicine, University of Iowa Carver College of Medicine , Iowa City, IA , United States of America
Huisman Henkjan
Electronic publication date: 2016 May 24
Publication date: 2016
Volume: 4
Electronic Location ID: e2057
Received 2016 Feb 23; Accepted 2016 Apr 29
Copyright: ©2016 Fedorov et al.
Copyright year: 2016
Copyright holder: Fedorov et al.
License: This is an open access article distributed under the terms of the Creative Commons Attribution License, which permits unrestricted use, distribution, reproduction and adaptation in any medium and for any purpose provided that it is properly attributed. For attribution, the original author(s), title, publication source (PeerJ) and either DOI or URL of the article must be cited.
License URL: https://creativecommons.org/licenses/by/4.0/

Keywords: Quantitative imaging, Imaging biomarker, Imaging informatics, DICOM, PET/CT imaging, Head and neck cancer, Image analysis, Cancer imaging, Interoperability, Open science

Funding: National Institutes of Health, National Cancer Institute U24 CA180918 U01 CA140206 U01 CA151261 National Institutes of Health (NIH) Clinical and Translational Science Award (CTSA) program U54 TR001356 This project was supported in part by funding from National Institutes of Health, National Cancer Institute, through grants U24 CA180918 (Quantitative Image Informatics for Cancer Research, QIICR), U01 CA140206, and U01 CA151261. Part of the infrastructure and services used at the University of Iowa are supported by the National Institutes of Health (NIH) Clinical and Translational Science Award (CTSA) program, grant U54 TR001356. The funders had no role in study design, data collection and analysis, decision to publish, or preparation of the manuscript.

==============================
Background. Imaging biomarkers hold tremendous promise for precision medicine clinical applications. Development of such biomarkers relies heavily on image post-processing tools for automated image quantitation. Their deployment in the context of clinical research necessitates interoperability with the clinical systems. Comparison with the established outcomes and evaluation tasks motivate integration of the clinical and imaging data, and the use of standardized approaches to support annotation and sharing of the analysis results and semantics. We developed the methodology and tools to support these tasks in Positron Emission Tomography and Computed Tomography (PET/CT) quantitative imaging (QI) biomarker development applied to head and neck cancer (HNC) treatment response assessment, using the Digital Imaging and Communications in Medicine (DICOM®) international standard and free open-source software.

Methods. Quantitative analysis of PET/CT imaging data collected on patients undergoing treatment for HNC was conducted. Processing steps included Standardized Uptake Value (SUV) normalization of the images, segmentation of the tumor using manual and semi-automatic approaches, automatic segmentation of the reference regions, and extraction of the volumetric segmentation-based measurements. Suitable components of the DICOM standard were identified to model the various types of data produced by the analysis. A developer toolkit of conversion routines and an Application Programming Interface (API) were contributed and applied to create a standards-based representation of the data.

Results. DICOM Real World Value Mapping, Segmentation and Structured Reporting objects were utilized for standards-compliant representation of the PET/CT QI analysis results and relevant clinical data. A number of correction proposals to the standard were developed. The open-source DICOM toolkit (DCMTK) was improved to simplify the task of DICOM encoding by introducing new API abstractions. Conversion and visualization tools utilizing this toolkit were developed. The encoded objects were validated for consistency and interoperability. The resulting dataset was deposited in the QIN-HEADNECK collection of The Cancer Imaging Archive (TCIA). Supporting tools for data analysis and DICOM conversion were made available as free open-source software.

Discussion. We presented a detailed investigation of the development and application of the DICOM model, as well as the supporting open-source tools and toolkits, to accommodate representation of the research data in QI biomarker development. We demonstrated that the DICOM standard can be used to represent the types of data relevant in HNC QI biomarker development, and encode their complex relationships. The resulting annotated objects are amenable to data mining applications, and are interoperable with a variety of systems that support the DICOM standard.

Introduction

Imaging has enormous untapped potential to improve clinical cancer treatment decision making. To harness this potential, research exploring the utility of image analysis to extract and process morphometric and functional biomarkers is essential. In the era of non-cytotoxic treatment agents, multi-modality image-guided therapies and rapidly evolving computational resources, quantitative imaging software performing such analyses can be transformative for precision medicine by enabling minimally invasive, objective and reproducible evaluation of image-based cancer treatment targeting and response. Post-processing algorithms are integral to high-throughput analysis and fine-grained differentiation of multiple molecular targets. Software tools used for such analyses must be robust and validated across a range of datasets collected for multiple subjects, acquisition devices, timepoints and institutions. Ensuring the validity of this software requires unambiguous specification of analysis protocols, documentation of the analysis results, and clear guidelines for their interpretation. Yet cancer research imaging data often does not exist in consistent formats that facilitate advancement of quantitative analysis. The infrastructure to support common data exchange and method sharing is lacking. These issues hinder development, validation and comparison of new approaches, secondary analysis and discovery of data, and comparison of results across sites and methodologies.

Recent initiatives such as the Quantitative Imaging Network (QIN) and Informatics Technology for Cancer Research (ITCR) of the National Cancer Institute (NCI), and Quantitative Imaging Biomarker Alliance (QIBA) of the Radiological Society of North America (RSNA) focus on a spectrum of issues related to quantitative imaging (QI) biomarker development, including both the validation and deployment of promising QI tools, and the development of the supporting infrastructure. Quantitative Image Informatics for Cancer Research (QIICR) is one of the projects of the ITCR consortium (http://qiicr.org, U01 CA190819). The overarching mission of QIICR is to provide free and open-source software (FOSS) QI analysis tools accompanied by the imaging data and analysis results stored in a standards-compliant structured fashion to support imaging biomarker development. Ultimately, our goal is to facilitate both the reuse of the shared research data and the acceleration of the translation of the QI methods and tools into clinical practice. QIICR is a collaboration with three sites of the NCI QIN (namely, Brigham and Women’s Hospital, University of Iowa, and Massachusetts General Hospital), each of which is focused on different diseases, and uses different imaging technologies and analysis methods. The research projects of interest at these three sites serve as use cases and testbeds for driving the requirements, testing and dissemination of the imaging informatics technology being developed by QIICR.

In this paper, we focus on one of the QIICR use cases—PET/CT QI biomarker development for treatment response in head and neck cancer (HNC)—to demonstrate how the use of the Digital Imaging and Communications in Medicine (DICOM®1 ) international standard (National Electrical Manufacturers Association (NEMA)), in conjunction with FOSS tools, can enable interoperable sharing of the quantitative imaging analysis results. The contributions of this work are twofold. First, we propose a DICOM-based approach to data sharing in QI research, and present the resulting dataset of clinical information and analysis results generated by a clinical research study investigating QI biomarkers in Positron Emission Tomography and Computed Tomography (PET/CT) imaging for predicting therapy outcome in the patients undergoing treatment for HNC. Second, we develop a suite of FOSS tools to facilitate encoding of the analysis results using the DICOM standard.

The research study that generated the data described in this work investigated the use of quantitation of the [18F]-fluorodeoxyglucose (FDG) tracer uptake in PET/CT images (CT is combined with PET for attenuation compensation as well as spatial localization). FDG PET/CT is commonly used for localization, characterization and qualitative assessment of therapy response in a variety of malignancies (Larson et al., 1999; Weber, 2006). Quantitative assessment of tumor burden using FDG PET/CT relies on a number of analysis steps, and can be sensitive to the processing technique and definition of the volumetric Region of Interest (ROI) (Boellaard, 2009; Vanderhoek, Perlman & Jeraj, 2012). A goal of the study that generated the data was to investigate the process of PET quantitation and propose improved ROI segmentation tools and a reproducible PET/CT quantitative analysis workflow. Steps involved in the analysis of the PET/CT images included normalization of the PET image data using the Standardized Uptake Value (SUV) body weight factor, segmentation of the tumor and involved lymph nodes using both manual and automated segmentation tools, segmentation of reference regions using automated segmentation tools, and quantitation of various statistics related to the tracer uptake from the segmented ROIs. The processing steps and their interactions are shown in Fig. 1, and are detailed in the Methods section.

Figure 1 Diagram of the interaction among the various data sources and processing steps that result in the dataset described in this paper.

Components of the dataset represented in DICOM are released publicly within The Cancer Imaging Archive (TCIA) QIN-HEADNECK collection. FOSS tools corresponding to the processing steps other than Reference Region (RR) segmentation (processing steps with the dashed outline) are available.

Most of the methods used for QI analysis that produced the data presented in this paper are accompanied by FOSS tools developed as part of the QIICR project. However, the main objective of this paper is not to discuss these analysis methods in detail, or to validate the tools implementing those analysis methods. Instead, we focus on the use of DICOM to enable structured, standardized, and interoperable communication of the annotated analysis results produced by those tools. Our goal is to facilitate access to the data and analysis results so other research groups can perform similar validation, compare the results to different methods or apply new tools to the imaging data.

The IEEE Standard Computer Dictionary defines interoperability as “the ability of two or more systems or components to exchange information and to use the information that has been exchanged” (ISO/IEC/IEEE, 2010). Interoperability implies the use of a common standard—ideally, an open standard—to engage the broad community of various stakeholders in industry and academia. We chose DICOM as the common standard, due to its broad and inclusive community of contributors, its ubiquitous adoption in the medical imaging domain and the suitability of its data model to accommodate the requirements of the use case. For rapidly evolving research applications like imaging biomarker development, it is also important to note that DICOM is a standard that is being continuously refined to address new community demands and technologies, while maintaining backwards compatibility with the existing user base. This process is enabled via the mechanism of Correction Proposals (CPs) and Supplements that can be submitted for consideration and review by the DICOM community, and are integrated into the standard through the formal process of discussion, refinement and voting.

DICOM is primarily used to support interoperability between clinical systems for image interchange (Haak et al., 2015). Consumption of the DICOM images produced by preclinical and clinical acquisition systems is widely supported in research tools, making an ever-growing stream of imaging data available to researchers. Sharing of results between different groups is widely regarded as a priority (Stodden, 2010; Walport & Brest, 2011; Boulton et al., 2011; Piwowar & Vision, 2013) and the failure to adopt standards for encoding results is flagged as a critical barrier (Chan et al., 2014). We argue (and demonstrate by example in this paper) that DICOM provides the means to support interchange of not only acquired images but also clinical data and various types of analysis results, with the goal of enabling their sharing and reuse. We recognize that the output of analysis results using DICOM is severely limited or non-existent in current tools. Instead, research tools often default to using locally defined or domain-specific formats (Kindlmann, 2004; Ibanez & Schroeder, 2005; Schroeder, Martin & Lorensen, 2006; NIfTI Data Format Working Group , 2005; MRC Cognition and Brain Sciences Unit, 2013), while commercial tools often limit export of the analysis results or utilize proprietary mechanisms. The research formats cover narrow use cases in restricted domains, ultimately compromising consistency, dissemination and reuse of the analysis results by fellow researchers. One of our objectives is to remedy this situation and provide the missing support of DICOM for QI applications in tools and toolkits.

Interoperable communication of analysis results between research and clinical systems is another critical consideration for validation and translation of QI precision medicine tools. The development and evaluation of research applications, data and software historically proceeds independently from clinical care and in distinct systems. Yet the extent to which data and software are interoperable between research and clinical environments directly impacts the ability to use clinical data for research, to use research applications in experimental clinical care, and to then translate research developments into clinical practice. Many barriers to such “translational” scenarios have been identified, among them being failure to use standard models and encoding formats in research and clinical environments (Katzan & Rudick, 2012; Chan et al., 2014) and failure to use standard codes (McDonald, Vreeman & Abhyankar, 2013).

Recent publications demonstrate that there is an increased recognition of the value of at least exporting images that are the result of research processing applications in DICOM format, so that they can be used to support various activities essential for imaging biomarker development (Krishnaraj et al., 2014). Such activities include consistently “tagging” analysis results to compare analyses done at different centers on different cohorts using different analysis tools (Waterton & Pylkkanen, 2012), supporting archival and distribution of the analysis results in a manner that enables indexing and secondary analysis (Chan et al., 2014) and transfer to and visualization of the analysis results in clinical systems in which metadata for patient identification and study management is required (Clark et al., 2013; Moore et al., 2015).

DICOM relies on coded terminology (Bidgood, 1997), both from standard external lexicons (such as the Systematized Nomenclature of Medicine (SNOMED®2) ) (Bidgood, 1998) as well as from the DICOM lexicon (National Electrical Manufacturers Association (NEMA), 2016a) when no suitable external terms are available (Bidgood et al., 1997). The “semantic” approach of using standard codes allows for greater reuse and harmonization with other data sets, since the need for natural language parsing of plain text during “data mining” is obviated by the commonality of standard codes for standard entities, such as anatomical regions, types of tumor, etc. (in the same manner as the “semantic web” (“Semantic Web—W3C, 2015”)).

The arguments presented above for the benefits of open standards such as DICOM are widely accepted, however adoption of such standards is not without effort. The DICOM standard is widely and fairly regarded in the research community as being non-trivial in complexity, while its documentation is extensive and difficult to navigate. Support of DICOM in toolkits is widespread, but mostly limited to the lower-level abstractions and more commonly oriented towards consuming rather than producing DICOM objects.3 Reference implementations and sample datasets illustrating the application of the certain parts of the standard are often absent. As with any complex endeavor, the DICOM standard itself is not without errors and may contain internal contradictions. The standard does not have (and does not claim to have) all of the features that are needed to support new or uncommon research use cases. These are some of the real obstacles for adoption of DICOM for communicating analysis results, both among manufacturers of commercial imaging workstations and within the QI research community.

In this contribution we take a number of steps to rectify this situation. We demonstrate the application of the DICOM standard to model and share a real example of a complex research dataset. We accompany this demonstration with the resulting dataset, source code of the conversion tools we used, developer toolkit and Application Program Interface (API) that we used to develop the conversion tools, and integrated user-level analysis and visualization tools, all available as FOSS. We provide detailed explanation of, and motivation for, using specific parts of the standard. Finally, we demonstrate how the standard itself can be improved via the community review process, to address errors and limitations, which can best be identified and solved by applying the standard to a real use case.

Materials and Methods

Patient cohort selection

The primary data was extracted from HNC patients with squamous cell carcinoma, all treated according to the standard of care at the University of Iowa Hospital and Clinics. Clinical practice was to obtain a FDG PET/CT for staging (prior to treatment) and then a second FDG PET/CT scan for response assessment at approximately three months following the completion of the initial therapy. All patient data was collected in compliance with HIPAA regulations under approval granted by the internal review board of the University of Iowa, approval #200503706. Written consent was obtained from the study participants. The imaging studies were acquired between 2004 and 2013. Patients that had a baseline and at least one post-therapy follow-up PET/CT were included in the research study. Patients were followed clinically and outcomes were available with a minimum of 2 years of follow-up. Patients may have had additional imaging studies following the three month response assessment FDG PET/CT based on clinical judgment and findings.

Clinical metadata for each patient was manually extracted from the electronic health records and included sex, age, smoking, and drinking history as well as pathology, stage, primary site location, and detailed location of involved nodal sites. Treatment details (e.g., radiation dose, technique, surgical intervention and chemotherapy delivery) and disease status and recurrences were recorded. All clinical metadata was de-identified and stored in a Postgres relational database locally. Measurements made on images that were used for clinical purposes and stored in the clinical records were not used during the conversion process, since new measurements were to be made, and homogeneity and accuracy of the clinical measurements could not be easily verified.

A total of 156 patients were identified as eligible for the study, with at least one PET/CT scan and related clinical data available for study (mean 3.05 studies/patient collected during a total of 472 visits). Fifty-nine patients from the cohort were processed using the methodology described in the following text. In one of those 59 patients both pre- and post-treatment imaging studies were processed, while in the rest of the patients only the baseline scan was analyzed.

Image acquisition

Pertinent details related to image acquisition such as reconstruction procedure, image resolution and injected dose were encoded in the DICOM image metadata by the scanner. After initial de-identification, the image data was stored in an eXtensible Neuroimaging Archive Toolkit (XNAT) (Marcus et al., 2007) local research archive at the University of Iowa.

Image processing

SUV is commonly utilized for a simple semi-quantitative analysis of PET images (Lucignani, Paganelli & Bombardieri, 2004). SUV Body Weight (SUVbw) is defined as the ratio of activity in tissue divided by the decay-corrected activity injected to the patient, normalized by body weight: SUVbw = (tissue activity)/(injected activity/weight). Several alternatives to SUWbw approach have been investigated including body surface area corrected (SUVbsa) and lean body mass corrected (SUVlbm) (Graham, Peterson & Hayward, 2000), but SUVbw remains the most commonly utilized quantity.

There are several underlying assumptions made in using FDG SUVs for measuring metabolic activity in lesions, such as accurate measurement of injected dose and accurate decay correction of all measurements (Graham, Peterson & Hayward, 2000). The failure of one or more of these assumptions can introduce variability in calculated SUVs. To mitigate this problem, an SUV ratio (SUVr) can be used, which represents the ratio of the SUV of a lesion to the SUV of a normal tissue Reference Region (RR) defined in the same acquisition.

In the project generating the data presented here, the primary cancer site and all involved lymph nodes were segmented separately to allow quantification of SUV for either the primary cancer site alone, total tumor burden, or on a per-region basis. Segmentation of the primary tumor and lymph nodes was done using two interactive segmentation tools within 3D Slicer (Fedorov et al., 2012). The first tool is a manual contouring tool, requiring the user to draw the boundary of a lesion on every slice using the Editor module of 3D Slicer. The second tool is semi-automated, performing segmentation in 3D using a specialized algorithm for segmenting HNC in FDG PET images, which is described and evaluated in detail in Beichel et al. (2016). This semi-automated segmentation approach treats the segmentation task as a graph-based optimization problem based on the ideas introduced by Yin et al. (2010). Starting with a user-provided approximate lesion center point, a graph structure is constructed in a local neighborhood, and a suitable cost function is derived based on local image statistics. A maximum flow algorithm is used for optimization. The resulting segmentation is converted from a graph-based representation to a labeled volume. To handle frequently occurring situations that are ambiguous, several segmentation modes are introduced to adapt the behavior of the base algorithm accordingly. In addition, “just enough interaction” based approaches are provided to enable the user to efficiently perform local and/or global refinement of initial segmentations. This semi-automated segmentation method is implemented in the PET Tumor Segmentation extension of 3D Slicer (QIICR, 2015a).

Since both manual and semi-automatic methods for HNC segmentation depend on user input, results are expected to be subject to intra- and inter-operator variation. To allow assessment of the impact of such variation on subsequent processing steps, each data set was reviewed and segmented using both methods by three readers, who were experts in HNC PET/CT image interpretation. Images were presented to the readers in random order. For each combination of the segmentation tool and reader, this process was performed twice, resulting in twelve segmentation sessions per patient. RRs in liver, cerebellum, and aortic arch were segmented automatically using the approach we presented earlier (Bauer et al., 2012).

Given the segmentations of the primary tumor and lymph nodes, a total of 22 quantitative indices were extracted from each of these regions using the PET-IndiC extension of 3D Slicer (QIICR, 2015b). The calculated quantitative indices consist of commonly utilized PET-specific indices such as maximum, mean and peak SUV (Wahl et al., 2009) and Total Lesion Glycolysis (TLG), as well as common summary statistics, which included median, variance and Root Mean Square (RMS) of SUV, and segmentation volume. Mean, maximum, minimum, standard deviation, median, and first and third quartiles were calculated for RRs.

Data modeling and conversion into DICOM representation

The DICOM standard provides a variety of objects that can be used to communicate information derived from the images. Regardless of the specific object type, DICOM requires that all objects contain so-called “composite context.” At the patient level, the composite context includes identifying and descriptive attributes such as patient name, ID, age and sex. The study context includes the date and time that the imaging study started, unique identification of the study and other information common to all series in the study. The composite context enables consistent indexing and cross-referencing of the various objects. In addition to shared composite context, derived DICOM objects typically contain explicit references to the “source” objects from which they were derived, which supports recording of the provenance of the object derivation as well as application functionality such as superimposition during rendering. Various relationships between the objects used in this study are shown in Fig. 2.

Figure 2 An illustration of the relationships among the DICOM objects discussed in this manuscript.

DICOM PET/CT is the original dataset obtained by the imaging equipment and is modified only by the de-identification procedure. DICOM RWVM, SEG, and measurement SR are derived objects. DICOM SR with the clinical information encodes the information about the patient originally stored in the relational database. Solid lines denote explicit reference of the object instances by the derived objects (referenced instance is pointed to by the arrow). Dashed bidirectional arrows denote commonality of identifiers (i.e., common composite context, e.g., at the Patient and Study level).

In the following sections we discuss the motivation for the choice of specific DICOM objects. We start with the PET and CT objects, since they were produced by the acquisition equipment and underwent only minor editing for de-identification. Next we describe the objects containing patient clinical data (clinical history and outcomes). Then we cover the derived imaging objects from simple to more complex:

∙ DICOM Real-World Value Mapping (RWVM) objects encode mapping of the image-specific SUV factor that is needed for normalization of the images and subsequent processing;

∙ DICOM Segmentation (SEG) objects encode labeling of the PET and CT image voxels into anatomical structures, such as primary tumor and liver ROI;

∙ DICOM Structured Reporting (SR) objects encode various measurements computed from the segmentation-defined regions on the normalized PET image volumes.

We follow the general pattern of discussing the scope and capabilities of the object at a high level, followed by an abbreviated summary of the design decisions made to meet the requirements of our use case. The reader is referred to the preprint version of this article (Fedorov et al., 2015a) for further discussions, which has been omitted for brevity. Corrections to DICOM that resulted directly from our experience are also listed. A separate section covers the implementation details of converting research representations into the DICOM format, and references the tools we developed for this purpose.

PET/CT image data

PET and CT image data were stored in the DICOM Positron Emission Tomography Image (National Electrical Manufacturers Association (NEMA), 2016b) and Computed Tomography Image (National Electrical Manufacturers Association (NEMA), 2016c), respectively. The image data obtained from the scanner was de-identified using a modified version of the Basic Attribute Confidentiality Profile defined by the DICOM in PS3.15 Appendix E.2 (National Electrical Manufacturers Association (NEMA), 2016d). Image de-identification was performed following the standard operational procedures established by The Cancer Imaging Archive (TCIA) (Clark et al., 2013; Moore et al., 2015). Research identifiers of the form QIN-HEADNECK-01-nnnn were assigned in place of the patient names and medical record numbers. Dates were shifted by the same fixed offset across all the datasets to maintain temporal relationships of the datasets. The de-identified images were then used for the remainder of the project (i.e., to make the measurements and convert them into derived DICOM objects), in order to mitigate the risk of leakage of patient identifiers into the publicly accessible analysis results.

Clinical information: DICOM SR

Relevant clinical information available for the subjects enrolled in the study included clinical history (such as the diagnosis and pathology, surgery and radiotherapy administration, and demographics) and outcomes (follow-up date and status, and the date of death, when applicable). This information is important for the interpretation and secondary reuse of the image and quantitative data set, since it contains the clinically relevant end-points for the evaluation of the biomarker performance, and it provides non-imaging predictors that can be used for machine learning. The clinical information was extracted from the operational Postgres research database, and retrospectively encoded in DICOM SR, one SR object per patient. The choice of DICOM SR for encoding clinical information is explained in Appendix S1.

DICOM SR objects (sometimes referred to as SR “documents”) contain information organized as a hierarchical content “tree” consisting of content items (tree nodes) (Clunie, 2000). These content items include containers, textual information, codes describing concept names (we will use “term” and “concept” interchangeably in this document) and values (where appropriate), references to images, and numeric values (National Electrical Manufacturers Association (NEMA), 2016e). DICOM SR templates define a pattern of content items and their relationships, constraining the general infrastructure for specific use cases (National Electrical Manufacturers Association (NEMA), 2016f). Each SR template is assigned a Template Identifier (TID). Templates may define the entire content of an object (i.e., be a “root” template) or may be a reusable common pattern of nested content to be included by higher level templates (i.e., be a “subordinate” template).

Each content item, except for those that are containers, can be thought of as a “name-value pair” (or alternatively, as a “question” and an “answer”). Containers can be considered “section headings,” and are often explicitly used as such when rendered in human-readable form. The top level (root node) of the content tree is always a container, and its name (concept) is often referred to as the “document title.” The concept name of a container or name-value pair (mandatory in most cases) is always coded using a code from a controlled terminology. The value may or may not be coded depending on the value type.

The use of controlled terminology is fundamental to DICOM SR. DICOM SR codes are defined as triplets of code value, coding scheme designator and code meaning (e.g., (F-02573, SRT, “Alcohol consumption”), where “SRT” is the DICOM designation for the SNOMED coding scheme). While DICOM allows for reuse of the codes defined in other terminologies, such as SNOMED, as well as those defined in the DICOM standard itself, so called “private” codes can also be defined by the creator of the object, when no standard codes are available. Such private codes are distinguished by a coding scheme designator that starts with a “99” prefix. The use of predefined codes not only provides semantic information, but also simplifies validation of the resulting objects. The codes that are allowed are constrained by the template. The constraints for values may be defined in the template itself, or in a “value set,” which in DICOM is called a Context Group (and labeled with a Context Group ID (CID)).

Though DICOM contains templates for clinical data for a few specific applications (e.g., cardiovascular (National Electrical Manufacturers Association (NEMA), 2016g) and breast (National Electrical Manufacturers Association (NEMA), 2016h)), it does not define a template to represent all the clinical data items of interest in our HNC QI research use case. Given the lack of a suitable standard template to represent this data, we developed our own set of custom templates for communicating the clinical information. In DICOM, such custom templates are referred to as “private templates,” even though they may be publicly shared and are required to be documented in the DICOM conformance statement of the product. These templates included information about biopsy, treatment and other relevant data. The relationships between the private templates are shown in Fig. 3, with a detailed description provided in Appendix S2. These templates follow the patterns of existing DICOM templates, with the intent that they might form the basis for future enhancements of the standard.

Figure 3 Relationships of the private DICOM SR templates used for encoding of the clinical information.

The top-level Clinical Data Report template incorporates subordinate templates, described in detail in Appendix S2.

No structured terminology was used at the time of initial clinical data collection, so terms with codes were selected retrospectively at the time of conversion of the data to DICOM SR. Our approach for selecting codes leveraged SNOMED (Cornet & De Keizer, 2008) and UMLS (Bodenreider, 2004) terminology as much as possible. The few concepts that could not be located in the SNOMED, UMLS or DICOM terminologies were added to a private coding scheme. All of the codes that are of relevance to this project are listed in Appendix S2.

Standardized uptake value: DICOM RWVM

The DICOM Real World Value Mapping (RWVM) object provides a mechanism to describe the calculation that was used (and can be reused) to create “real world values” (such as SUV) from stored pixel data values. A RWVM can be embedded within another DICOM object (such as an acquired or derived image), or it can be encoded as a standalone object (National Electrical Manufacturers Association (NEMA), 2016i), which in turn can either be referenced from other objects, such as SRs, or recognized as being relevant from the commonality of patient and study identifiers.

We chose to create a standalone RWVM object to encode SUVbw factor and leave the original (de-identified) activity-concentration images unchanged. The RWVM object encodes the scale factor, the range of stored pixel values to which it applies, and standard codes that specify the quantity that the scaled (real world) value represents (in this case, the SUV), the measurement method (the SUV body weight calculation method) and the measurement units (g/ml{SUVbw}). The DCM coding scheme is used for the quantity and the measurement method, and, as is the case throughout DICOM, the Unified Code for Units of Measure (UCUM) system (Schadow et al., 1999) is used for the units. The RWVM object also includes references to all of the PET image objects to which it applies.

The following corrections to the standard were proposed to remedy the errors or limitations of the standard identified while developing DICOM representation of the SUVbw factors for this project:

1. CP 13874 :addition of quantity descriptors to Real World Value Maps (applies to the 2014b version of the standard). The original definition of the RWVM in DICOM only defined the encoding of measurement units. We proposed an improvement to the standard to include the definition of quantity in the RWVM encoding.

2. CP 1392: addition of quantity descriptors and measurements for PET (applies to 2014b). This CP added new concepts related to encoding of the PET measurements that were missing in the standard, but were required by our use case.

Image segmentation: DICOM SEG

The imaging time point was defined as an ordinal number corresponding to the imaging study performed for the patient in the course of management of the specific condition, with time point 1 corresponding to the baseline/staging study. For each such time point we encoded segmentations prepared using image processing steps discussed earlier.

DICOM provides different mechanisms for encoding ROIs obtained by segmentation, as discussed in Fedorov et al. (2015a). The choice of the most suitable mechanism depends on the use case. Since the native representation of the segmentation results were labelled individual voxels, rather than a surface mesh or isocontours, we selected the DICOM Segmentation image (SEG) object as the most appropriate for encoding the ROIs.

The SEG objects were organized as follows, to be consistent with the pattern that would likely be used by tools that created them prospectively rather than retrospectively:

∙ Each of the RRs is stored as a separate object, since each of the RRs was segmented using a distinct automatic method, using data from different modalities (the aortic arch was segmented on the CT images, and the cerebellum and liver ROI were segmented on the PET images).

∙ The primary tumor and involved lymph nodes segmented for each combination of operator/segmentation method/session were stored together as different segments in a single object, since both the tumor and nodes were segmented during the same session, with the segmentation of one structure being identified while considering the neighboring structures.

∙ The identifier of the operator (reader) for the manual and semi-automated segmentation results was stored in the ContentCreatorName5 attribute.

∙ The identifier of the imaging time point was encoded as a positive integer, stored in the ClinicalTrialTimePointID attribute.

∙ The identifier of the segmentation session for primary tumor and lymph nodes was encoded in the ClinicalTrialSeriesID attribute.

∙ The type of algorithm used was encoded in the SegmentAlgorithmType attribute as MANUAL, SEMIAUTOMATIC or AUTOMATIC, as appropriate.

∙ The suggested color for each of the segmented structures was encoded in the RecommendedDisplayCIELabValue attribute.

The semantics of the segments were communicated using the standard AnatomicRegion (and its modifier in AnatomicRegionModifier sequence, when necessary), SegmentedPropertyType and SegmentedPropertyCategory sequences. For example, the semantics of a primary tumor was encoded as follows:

Segmented Property Category = (M-01000, SRT, "Morphologically Altered Structure") Segmented Property Type = (M-80003, SRT, "Neoplasm, Primary") Anatomic Region = (T-53131, SRT, "base of tongue")

DICOM defines a relatively small set of segmentation property categories, listed in CID 7150 (National Electrical Manufacturers Association (NEMA), 2016j), and a considerably larger set of segmentation property types in CID 7151 (National Electrical Manufacturers Association (NEMA), 2016k). There is no direct relationship specified in the standard between category and type, and the choice of an appropriate category is left to the discretion of the implementer (arguably the standard could be improved by grouping the types and assigning them to, and requiring them for, specific categories).

Sometimes segmentations are performed for purely anatomical reasons (e.g., for anatomical atlases), in which case there is no meaningful additional property type to record. In such cases, the anatomy is encoded directly in SegmentedPropertyType, without the need for a separate AnatomicRegionSequence. In other cases, segmentations are performed that apply to anatomical structures, but which segment them into different types of tissue. In these cases, the SegmentedPropertyType is used to encode the type of tissue (e.g., primary tumor, secondary tumor, necrosis) and the AnatomicRegionSequence can be used to encode the anatomic location (e.g., which organ, group of lymph nodes, etc.). Sometimes the anatomy is irrelevant and not encoded at all, and the SegmentedPropertyType just encodes the type of material segmented. This distinction was clarified by the authors in an earlier DICOM correction proposal CP 1258. In this project we are encoding both the nature (category and type) of the segmented area and its anatomic location.

Lymph nodes are encoded similarly, but with only the general region (head and neck) recorded rather than a precise code for the lymph node group, because of the lack of the detailed information about the specific lymph node name in the original dataset due to practical difficulties in assigning such a precise name when segmentation was performed:

Segmented Property Category = (M-01000, SRT, "Morphologically Altered Structure") Segmented Property Type = (M-80006, SRT, "Neoplasm, Secondary") Anatomic Region = (T-C4004, SRT, "lymph node of head and neck")

Semantics of the RR segmentations are communicated using the “spatial relationship concept” category:

Segmented Property Category = (R-42018, SRT, "Spatial and Relational Concept") Segmented Property Type = (C94970, NCIt, "Reference Region") Anatomic Region = (T-62000, SRT, "Liver")

Binary segmentations are encoded in the PixelData attribute of the SEG object, and are represented as a contiguous array of bits, with one bit per voxel for each frame. There are separate frames for each slice of the volume, though all are encoded in a single multi-frame object. When multiple segments (i.e., primary tumor and lymph nodes) are produced by the operator during a single session using a single segmentation tool, they are stored in a single SEG object, with each segment for each slice stored in a separate frame. Empty frames that do not contain any voxels of the segmentation are elided, to reduce the size of the encoded objects. The matrix size (rows and columns) is not abbreviated to a rectangular bounding box enclosing the region of interest, which would be a further possible object size optimization (i.e., each frame has the dimensions of the original image). Similar to the RWVM objects, SEG objects include references to the SOP Instance UIDs of the images (slices) that were segmented.

The process of encoding the segmented ROIs in DICOM led to the development of the following correction proposals:

1. CP 1406: add codes for tumor sites (applies to 2014c). The uncoded (plain text) labels of all the tumor regions used in this project were analyzed to identify common terms that were then mapped to SNOMED concepts. The resulting terms were introduced into the DICOM standard in the form of new context groups for lymph nodes (CID 7600) and HNC anatomic sites (CID 7601). A distinction between concepts for primary and secondary neoplasms was introduced in the same proposal.

2. CP 1426: correct condition in pixel measures, plane position and orientation functional groups for segmentation (applies to 2015a). Prior to this correction, the presence of the essential attributes that are needed for volumetric reconstruction of the segmentation image volumes was conditioned on attributes that were optional or not defined in segmentation objects.

3. CP 1464: add reference region segmentation property type (applies to 2015c). This correction added the codes needed to describe RRs, using the NCI Thesaurus terminology.

4. CP 1496: add tracking identifier and UID to segmentation instances (applies to 2015c). Use of a common Tracking UID allows to establish correspondence between segments encoded in various segmentation objects that represent the same region being segmented (i.e., across different time points, modalities, operators). Tracking UIDs were already present in the SR measurements objects, which can reference segmentation objects, but were not encoded directly in the segmentation objects themselves.

Quantitative measurements: DICOM SR

To encode the PET SUV ROI measurements in DICOM, we specified the terminology that defines the measurement quantities, modifiers and units for each measurement of interest needed. The vocabulary required was not specified in any single standard context group. Concepts from various standard context groups were therefore leveraged as appropriate. The strategy to find a suitable term was to first consult those already in DICOM, then search for related concepts in UMLS, SNOMED, and the NCI Thesaurus. If no existing concept could be found, we introduced a new code and definition in a private 99PMP coding scheme, while referencing a relevant publication, if available. All of the terms used are described in Appendix S3. The reader is referred to Fedorov et al., (2015a) for additional discussion of the selection of the quantity codes.

The measurements were encoded as DICOM SR objects using the standard root template TID 1500 defined in PS3.16 (National Electrical Manufacturers Association (NEMA), 2016l), which makes use of the sub-ordinate templates shown in Fig. 4. TID 1500 contains a preamble that describes general characteristics relevant to the measurement, such as an Image Library container (National Electrical Manufacturers Association (NEMA), 2016m), which lists the UIDs of the images in the original image series, radiopharmaceutical agent, and other items related to the acquisition protocol that may be relevant during interpretation. The Imaging Measurements container (section heading) includes the following attributes, which have special meaning in the context of our use case:

∙ Activity session: a positive integer that encoded the segmentation session by the operator.

∙ Tracking identifier: a human-readable identifier of the finding, which is not required to be unique. In our project, RRs had tracking identifiers coded as “referenceRegionName reference region,” where referenceRegionName was one of “liver,” “cerebellum” or “aortic arch.” The primary tumor identifier was always set to “primary tumor,” individual lymph nodes were identified as “lymph node nodeID,” where nodeID is a positive integer. As mentioned earlier, lymph nodes were not tracked (i.e., their nodeID did not identify the specific lymph node across time points or reading sessions).

∙ Tracking unique identifier: a DICOM standard UUID-derived (random) identifier with a “2.25.” prefix (National Electrical Manufacturers Association (NEMA), 2016n): a primary lesion unique identifier that was used to track the lesion and reference regions across time points.

∙ Time point: a positive integer that encoded the temporal order of the imaging study within the course of management of the given patient.

∙ Referenced segment and source series for image segmentation: the identifiers of the segment and the segmentation object representing the ROI used in the measurement group, and the identifier of the series that was segmented.

∙ Finding site: the coded anatomical location of the finding.

Related groups of measurements were encoded as a list, preceded by the codes of one or more findings, following the structure defined by TID 1411 Volumetric ROI Measurements (National Electrical Manufacturers Association (NEMA), 2016o), which in turn invokes TID 1419 ROI Measurements (National Electrical Manufacturers Association (NEMA), 2016p), as summarized in Fig. 4. Each group of measurements was derived from the ROIs that applied to the voxels of a single reconstruction of a PET acquisition (image series). One SR measurement object was created for each SEG object. Voxels in the ROI used for the derivation of the measurements were encoded as one segment of a SEG object. Both the SEG image objects and the segment number used by the derivation were referenced for each measurement group in the SR. An example of the structure of the Imaging Measurements is presented in Appendix S4.

Figure 4 The family of DICOM SR templates used for communicating the PET measurements.

All of the templates used to encode derived measurements are included in the DICOM standard.

The following DICOM standard corrections were contributed while developing the conversion methodology:

1. CP 1366: correction of relationships in planar and volumetric ROI templates (applies to 2014b). In the process of data encoding, we identified errors in the definition of the relationships in some templates.

2. CP 1386: addition of measurement report root template for planar and volumetric ROIs (applies to 2014b). Before the introduction of this root template, measurement templates could only be used to construct subordinate objects included in other templates, but not to encode standalone measurement objects. This CP also added some of the codes needed for this project, and allowed common content items to be factored out of individual measurements to the group level.

3. CP 1388: add real world value map reference to measurements (applies to 2014b). This CP added an explicit reference to the RWVM instance that was used to calculate the measurements to the measurements SR object template.

4. CP 1389: factor common descriptions out of image library entries (applies to 2014b). We introduced simplifications to the structure of the measurements SR object by allowing a group of images to share common image library attributes, greatly reducing the size and improving the readability of the object in cases when measurements were derived from many single frame images.

5. CP 1465: add type of finding to measurement SR templates (applies to 2015c). The measurement template was amended to include the type of finding, which is distinct from its anatomical location.

6. CP 1466: add session to measurements group (applies to 2015c). An extra item was added to the measurement template to enable encoding of the session identifier to support experiments where the measurement of the same finding is performed several times in order to evaluate its repeatability.

Implementation of the Conversion to DICOM Format

Our overall strategy for data conversion was developed to accommodate the organization of the data at the site conducting the study. Customized routines were developed to perform conversion of the individual components of the data stored in the internal databases. SUV normalization and quantitative measurements were calculated using the FOSS tools developed as part of this project. Segmentations were converted from the results obtained before the open-source implementation of the semi-automatic segmentation tools was released. The top-level script that was used to perform the conversion of a complete dataset by invoking conversion routines for the individual data types is available in the Iowa2DICOM code repository (QIICR, 2015c).

Clinical information: DICOM SR

Clinical data was exported from the internal SQL database as a tab-delimited text file. An XSLT script was used to convert the tab-delimited representation into XML form, followed by another XSLT transformation that produced an XML representation of an object that follows DICOM SR template TID QIICR_2000 documented in Appendix S2. Finally, the resulting XML representation was converted into DICOM format using existing functionality of the PixelMed toolkit (Clunie, 2015a). The conversion scripts are available in a public source code repository (QIICR, 2015d). The DICOM series containing the clinical data DICOM SR were assigned to a study separate from the one for the imaging and derived data, with both the StudyDescription and SeriesDescription attribute set to “Clinical Data.”

Standardized uptake value: DICOM RWVM

RWVM objects were generated in batch mode using the SUV calculation plugin of 3D Slicer (QIICR, 2015e). The plugin operated on the list of files corresponding to the PET series DICOM objects, calculated SUVbw factor and produced a single RWVM object. Injected dose, patient weight, radionuclide half-life and injection time were obtained from the DICOM PET image header.

Image segmentation: DICOM SEG

The process of converting segmentation results into DICOM representation was facilitated by the FOSS DICOM software library implementation available in DCMTK (DICOM Toolkit) and maintained by OFFIS in Germany (Eichelberg et al., 2004). To simplify the task of creating SEG objects for this project and other similar efforts, we extended DCMTK with three new libraries, which are now included in the official distribution of DCMTK: dcmiod, dcmfg and dcmseg (Fedorov et al., 2015a). The conversion was performed using batch mode tools SEG2NRRD (conversion from DICOM SEG to NRRD research format) and EncodeSEG (conversion from research segmentation format to DICOM SEG). These tools are included in the Iowa2DICOM repository referenced above.

Volumetric measurements: DICOM SR

The process of calculation and encoding of the ROI measurements was implemented in 2 steps. First, measurements of interest were calculated in batch mode using the QuantitativeIndicesCLI tool available within PET-IndiC extension of 3D Slicer (QIICR, 2015b). The tool accepted the SUV-normalized image volume and the segmentation label saved using a domain-specific format, such as NRRD or NIfTI, and produced a text file encoding the measurements as key-value pairs. The keys of the output correspond to the research labels assigned to the measurement classes. Not all of the measurements were generated for each of the ROIs. Specifically, calculation of a meaningful value for SUV peak (Wahl et al., 2009) was not possible when the ROI was too small. In the cases when the measurement was not generated by the tool, it was omitted from the DICOM SR measurements object.

Next, we used EncodeMeasurementsSR converter available within the Iowa2DICOM repository (QIICR, 2015c) to generate DICOM SR objects containing the calculated measurements. This converter accepted as input the list of DICOM PET object file names, the SEG object file name, and the text measurements, and produced the DICOM SR object according to TID 1500. The conversion utilized the dcmsr library of DCMTK, which provided interfaces to create and iterate through a tree of DICOM SR object content.

Validation of DICOM encoded objects

The dciodvfy tool (Clunie, 2015b) was used to ensure that an object complied with the basic DICOM encoding rules and contained the appropriate required attributes for the images, SEG, RWVM, and SR objects. This tool did not validate compliance with specific SR templates, only that valid combinations of content items and relationships were present.

The dcentvfy tool was used to validate that a set of DICOM objects contained the correct values for all attributes for the same entity level in the DICOM Information Model (i.e., that all patient attributes were the same for the objects with the same PatientID value, that all study attributes were the same for objects with the same StudyInstanceUID value, etc.). This tool was particularly helpful when objects were created along different paths or by using different tools than the original images, and/or uploaded to the distribution archive on separate occasions.

The com.pixelmed.validate.DicomSRValidator tool (Clunie, 2015c) was applied to validate compliance with the subset of SR templates that were supported by the tool, which included the TID 1500 root template and the subordinate templates used in this project. The validation consisted of checking that the required content items were present at the correct level in the content tree, that conditional content items were present when specified conditions were satisfied, that correct concepts and required values from specified context groups were used, and that concepts were encoded with the expected code meanings. Warnings were triggered when unrecognized content items were detected (which often signaled that a content item had been misplaced in the tree).

Code availability

All of the code, with the exception of that for the automatic segmentation of PET RRs, is available as FOSS without any restrictions on its use. Specifically, we share the following FOSS tools used for PET/CT data analysis:

1. 3D Slicer (Fedorov et al., 2012) was used as the platform for implementation of all the processing tools. Home page: http://slicer.org. Source code: http://github.org/Slicer/Slicer.

2. PET SUV conversion: 3D Slicer PETDICOMExtension extension. Home page: http://wiki.slicer.org/slicerWiki/index.php/Documentation/Nightly/Modules/DICOMPETSUVPlugin. Source code: https://github.com/QIICR/Slicer-PETDICOMExtension/tree/master/DICOMPETSUVPlugin.

3. Manual PET segmentation: Editor module of 3D Slicer (documentation and source code URLs are as above for 3D Slicer).

4. Semi-automated PET segmentation: PETTumorSegmentation extension. Home page: http://wiki.slicer.org/slicerWiki/index.php/Documentation/Nightly/Extensions/PETTumorSegmentation. Source code: https://github.com/QIICR/PETTumorSegmentation.

5. PET quantitative index calculation: 3D Slicer PET-IndiC extension. Home page: http://wiki.slicer.org/slicerWiki/index.php/Documentation/Nightly/Extensions/PET-IndiC. Source code: https://github.com/QIICR/PET-IndiC.

In addition to the image processing tools listed above, we provide source code of the FOSS tools used to create DICOM representations of the analysis results in the Iowa2DICOM repository: https://github.com/QIICR/Iowa2DICOM.

Results

Clinical data and the analysis results for the total of 60 PET/CT imaging studies were encoded in the DICOM format using the procedures described. One patient had a repeat imaging study. The remainder had only the baseline study augmented with the clinical data and quantitative analysis results DICOM objects.

One RWVM object, 15 SEG objects (3 RRs and tumor/lymph nodes segmentations by 3 readers using 2 tools during 2 reading sessions), and 15 volumetric measurement SR objects (one per SEG) were produced for each imaging study.

The DICOM objects were added to the QIN-HEADNECK collection of TCIA (The Cancer Imaging Archive (TCIA), 2015) and are available for public access.6 TCIA was selected for archival of the resulting data since it was capable of storing and indexing the DICOM objects used, and was (and still is) the QIN-recommended data sharing platform, and the analysis generating the encoded data was done as part of the QIN activities at the University of Iowa.

Standalone validation and consistency checks were conducted as described above. In addition, interoperability testing was performed as described in the remainder of this section to confirm that the objects could be ingested and used by commonly available tools and toolkits: DCMTK (OFFIS, 2014), GDCM (Malaterre, 2015), dicom3tools (Clunie, 2009) and PixelMed (Clunie, 2015a).

The traditional DICOM encoding format is binary, and data stored in that form is most easily visualized after transformation into a human-readable text format, for which different options exist. One commonly used approach is to look at a so-called “dump,” which lists each attribute with its tag, type (value representation), name and value (with hierarchical nesting of sequences shown as required). The following publicly available tools were tested and able to successfully dump the objects we created:

∙ DCMTK dcmdump (dump2dcm for reverse conversion)

∙ GDCM gdcmdump

∙ dicom3tools dcdump

∙ PixelMed com.pixelmed.dicom.AttributeList

∙ PixelMed com.pixelmed.dicom.AttributeTreeBrowser

It is also possible to convert the DICOM format into XML or JSON representations, either according to schemas recently defined by the DICOM standard for this purpose (National Electrical Manufacturers Association (NEMA), 2015), or using non-standard schemas. These representations make the data amenable for consumption by the variety of established tools such as various NoSQL databases, XML query and transformation engines, etc., and are also nominally “human-readable.” We tested the following tools to confirm they could perform conversion of the objects we generated into an XML representation:

∙ DCMTK dcm2xml (xml2dcm for reverse conversion)

∙ GDCM gdcmxml

∙ PixelMed com.pixelmed.dicom.XMLRepresentationOfDicomObjectFactory

DICOM SR objects can also be interpreted at a higher level of abstraction, which describes the content items of the content tree instead of the individual attributes that compose each content item. Such SR content tree “dumps” are more amenable to human interpretation than the attribute level dumps. The following tools were tested to produce SR tree dumps of the objects we generated:

∙ DCMTK dsrdump

∙ dicom3tools dcsrdump

∙ PixelMed com.pixelmed.dicom.StructuredReportBrowser

DICOM SR objects can also be converted into an XML representation according to a schema defined at the level of abstraction of the SR content tree rather than the individual attribute level. Such representations are very suitable for integration of the DICOM data with a variety of XML-oriented tools. A caveat is that DICOM has not yet established a standard schema for such a conversion, so the XML representation is dependent on the schema implemented by the specific tool. The following tools were tested and found to be capable of generating XML representations of the DICOM SR content for the objects we generated:

∙ DCMTK dsr2xml (xml2dsr for reverse conversion)

∙ PixelMed com.pixelmed.dicom.XMLRepresentationOfStructuredReportObjectFactory (bidirectional)

The DCMTK dsr2html tool can be used to generate an HTML representation of the SR content tree that can be rendered in a human-readable form in any HTML viewer. The dsr2html tool was tested and found to be able to render the SR objects that we generated.

Finally, we provide an XSLT transformation that can be used to convert a DICOM SR document that follows template TID 1500 into a comma-delimited CSV text (QIIR, 2016b).

All of the tools discussed above are command line tools. Interactive applications that wrap those command line tools are also available. The dcmjs dump (Common, 2015) tool provides a web interface to DCMTK dcmdump, with the data processing done fully on the client side. The dicom-dump package (QIICR, 2015f) of the FOSS Atom editor wraps both dcmdump and dsrdump tools of DCMTK, and can be used to interactively invoke those tools on the DICOM objects opened in the Atom editor.

To illustrate the various options for examining DICOM data, we provide examples of output in different forms for the tumor measurements SR object for subject QIN-HEADNECK-01-00024. At the level of DICOM attributes, measurement of the SUVbw peak is shown in Fig. 5. Another view of this same portion of the object in DCMTK-specific SR XML is shown in Fig. 6.

Figure 5 An attribute-level dump corresponding to the section of the DICOM SR measurements.

The text shown is an excerpt of the complete object dump encoding SUVbw peak value for subject QIN-HEADNECK-01-0024, series “tumor measurements—User1 Manual trial 1”, as displayed in the Atom editor using dicom-dump package.

Figure 6 An XML representation corresponding to the section of the DICOM SR measurements.

The excerpt shown is encoding SUVbw peak measurement for subject QIN-HEADNECK-01-0024, series “tumor measurements—User1 Manual trial 1.”

By comparison, an SR tree level text dump of the same content as produced by dsrdump appears as follows:

<contains NUM:(126401,DCM,"SUVbw") = "5.90721" ({SUVbw} g/ml,UCUM,"Standardized Uptake Value body weight")> <has concept mod CODE:(121401,DCM,"Derivation") = (126031,DCM,"Peak Value Within ROI")>

A rendered view of a section of the HTML representation of the same object as produced by dsr2html is shown in Fig. 7.

Figure 7 A rendered view of an HTML representation of the SR measurements object tree.

The content shown is for subject QIN-HEADNECK-01-0024, series “tumor measurements—User1 Manual trial 1,” as generated by the DCMTK dsr2html tool and rendered in a Chrome browser. SUVbw peak measurement is highlighted by the red rectangle.

The foregoing checks did not serve to test more complex application-level interoperability. Additional tests were performed for the SEG objects. Since ROIs encoded as segmentations may be visualized in relation to the images from which they were segmented, we investigated the interoperability of several imaging workstations with respect to their ability to correctly render segmentations superimposed on the PET images. The following software was tested:

∙ 3D Slicer (Reporting extension, starting from Nov 22, 2015 nightly build version)

∙ ePAD v1.7 (Stanford Medicine, 2015)

∙ AIM on ClearCanvas v4.6.0.3 (Mongkolwat, 2015)

∙ Brainlab PDM v2.2 (commercial workstation) (Brainlab AG, Feldkirchen, Germany)

Each of these platforms was capable of successfully importing the SEG objects and displaying the encoded segments. An example of the rendering of the segmentations in 3D Slicer is shown in Fig. 8.

Figure 8 Example of the segmentation results visualization initialized from DICOM representation.

Shown is subject QIN-HEADNECK-01-00024, as displayed in 3D Slicer software. The primary tumor is shown in green and the lymph node metastasis in yellow. (A): overlay of the secondary tumor outline in yellow over a coronal reformat of the SUV-normalized PET volume. (B): overlay of the secondary tumor outline and SUV-normalized PET volume thresholded to highlight the areas of uptake over a coronal reformat of the CT volume. (C): maximum intensity projection (MIP) view of the PET volume composed with the surface rendering of both the primary and secondary tumors.

Discussion

Realistic quantitative imaging research scenarios necessitate the use of a variety of data sources and processing routines, making the results of such analyses inherently complex. Our goal was to provide a complete and reproducible description of the process, both from the data modeling and implementation perspectives. A key strategy for mitigation of complexity is the provision of appropriate tools. We hope that the burden of complexity on the individual researcher can be minimized, whilst reusability and interoperability can be maximized, by leveraging and improving existing DICOM FOSS tools and toolkits, instrumenting widely used research applications with DICOM capability, and providing a clear path selecting and linking an appropriate, relevant, and sufficient subset of DICOM capabilities for the research use case.

We believe this work is the first to demonstrate the utility of the DICOM standard for interoperable quantitative result encoding in the QI research domain, complete with the publicly available FOSS implementing the conversion and interpretation/visualization tools, encoded objects and documentation describing the specialized templates used for data encoding. Furthermore, we intentionally described the details of the various correction proposals that were contributed to the standard in the course of our work, to demonstrate that DICOM is an evolving standard that is open to improvements as needed to support research use cases. The improvements to the standard contributed by this project have wider applicability and, we hope, will greatly simplify the task for adopters of the DICOM approach.

The tools available in the Iowa2DICOM repository were developed for the specific HNC QI use case presented in this manuscript. As such, the repository has served the intended purpose of producing the dataset described, and is not maintained. We provide the source code of Iowa2DICOM to facilitate reproducible research and to provide technical insight into our methods. The SEG converters can be used for general purposes and have since been incorporated in 3D Slicer to enable import and export of DICOM SEG objects. We are also working on the next iteration of the conversion tools in the new dcmqi (DICOM for QI) library (QIICR, 2016a) to provide general purpose DICOM conversion tools. Unlike Iowa2DICOM, which is dependent on 3D Slicer build tree, dcmqi is self-contained. It is under development and will be maintained by the QIICR project. As of writing, dcmqi incorporates the SEG conversion tools and includes basic examples, sample datasets and usage instructions.

Data conversion, as implemented and described in this paper, was performed retrospectively. We did not use DICOM as the operational data format, but instead adopted it to enable archival and sharing of the final analysis results, since the purpose was to reuse data already acquired for a research study to test the hypothesis, rather than wait until improved tools were fully deployed for prospective data acquisition. We are not arguing that retrospective conversion is preferred, quite the contrary. It is practical though, since historical analysis pipelines often contain tools developed using different toolkits and languages that may not yet have support for the various DICOM objects we utilized. The installed base of research tools may also not yet contain sufficient mechanisms for maintaining and propagating the patient and study level information (the composite context). Our project demonstrates how, in situations like the one encountered in this project, composite context can be recovered and merged into the shared results retrospectively, to re-associate acquired images, derived results and clinical data. Addressing this key barrier to interoperability with the clinical environment should be a high priority for the research community, particularly since scalability to large experiments and the conduct of clinical trials (especially those spread across multiple sites or using multiple tools), requires a solution to manage data identity and provenance. That said, the choice of format for interoperable exchange versus that for internal operational use can remain distinct to the extent deemed appropriate for any particular research scenario.

The work presented in this paper is a step towards improving support of quantitative imaging research use cases in DICOM, and improving support of the relevant parts of the DICOM standard in both FOSS and commercial tools and toolkits. We are actively engaged in improved integration of DCMTK with 3D Slicer to provide streamlined user interfaces that empower end users to store the results of their work as appropriate DICOM objects with minimum extra effort. Although the specific use case described in this paper involve PET/CT, the approach has broad applicability for interoperable communication of segmentation and quantitative analysis results independent of the imaging modality. At the level of developer toolkits, we have recently completed the implementation of an API in DCMTK to support abstractions related to the generation of volumetric measurement SR objects (TID 1500).7 We are also in the process of extending the DCMTK API to support the creation of Enhanced Multi-frame objects for Magnetic Resonance Imaging (MRI). We are planning to use that functionality for other QI biomarker use cases being investigated by QIICR that focus on the use of multiparametric MRI in glioblastoma and prostate cancer. To support those use cases that involve analyses that generate derived functional maps of tissue properties, QIICR has also contributed to the development of the Parametric Map object in DICOM (National Electrical Manufacturers Association (NEMA), 2016q), now part of the standard, which supports encoding of floating point pixel data without being restricted to rescaling of integer values, finally resolving a longstanding perceived weakness of DICOM for research applications.

Another area of QIICR focus is the development of tools to ease the process of interacting with the standard and exploring the content of DICOM data. In this area, we have developed an initial version of a DICOM search index that provides an alternative interface to explore the DICOM standard (QIICR, 2015g), and contributed the dicom-dump package to the popular Atom editor discussed earlier (QIICR, 2015f). These additional activities are intended to assist a diverse variety of groups, which include academic QI researchers (both technical and clinical), software developers implementing QI analysis tools, clinical end users, and developers of the commercial tools deploying QI biomarkers. Our goal is to make it easier for interested parties to explore, evaluate and implement DICOM capabilities relevant for QI research. We hope these efforts will contribute to the technical solution of the overarching problem of standardized and meaningful sharing of reproducible research results, as well as improve the integration of the research tools with clinical systems to facilitate the translation of QI biomarker clinical trials and clinical research studies into clinical practice.

Conclusions

We have presented a detailed investigation of the development and application of the DICOM standard and supporting FOSS tools to encode research data for quantitative imaging biomarker development. Using the real-life research scenario of HNC PET/CT quantitative image analysis, we demonstrated that the DICOM standard is capable of representing various types of analysis results and their interrelationships. The resulting data objects are annotated in a standard manner, and utilize consistent and widely used codes for communicating semantics. They are also interoperable with the variety of tools readily available to the researcher, as well as commercial clinical imaging and analysis systems (which universally support many aspects of the DICOM standard).

The work presented is a result of two years of activities of the QIICR project, but it builds upon the foundation established by the various research groups, communities and FOSS projects, such as 3D Slicer and DCMTK, decades before QIICR. We are committed to continue working with those groups and communities, as well as other stakeholders and adopters interested in remedying the status quo of very limited sharing of the quantitative image analysis results in the imaging community.

Supplemental Information

Appendix S1 Background and motivation for using DICOM to encode patient clinical information

Click here for additional data file.

Appendix S2 QIICR Iowa Head and Neck Clinical Data DICOM SR Template

Click here for additional data file.

Appendix S3 QIICR PET SUV measurements of interest and their corresponding codes

Click here for additional data file.

Appendix S4 Example of a populated TID 1500 imaging measurements container

Click here for additional data file.

We thank John Sunderland for the help with PET/CT image acquisition; Markus van Tol for his contribution to the implementation of the PET tumor segmentation module in 3D Slicer; Kirk Smith for his help in archiving the DICOM data on TCIA; Jean-Christophe Fillion-Robin, Andras Lasso, Nicole Aucoin, Christian Herz, and the 3D Slicer community for their contribution to the development of the relevant 3D Slicer functionality.

Evaluation of interoperability of the resulting DICOM segmentation objects with ePAD, AIM on ClearCanvas and Brainlab tools was performed as part of a Quantitative Imaging Reading Room exhibit at the 2015 convention of the Radiological Society of North America (RSNA) (Fedorov et al., 2015b). We thank Daniel Rubin, Pattanasak Mongkolwat and David Flade for providing access to and facilitating testing of the interoperability of the respective tools.

Additional Information and Declarations

Competing Interests

Author Contributions

Human Ethics

Data Availability

1 DICOM is the registered trademark of the National Electrical Manufacturers Association (NEMA) for its standards publications relating to digital communications of medical information.

2 SNOMED is a registered trademark of the International Health Terminology Standards Development Organisation (IHTSDO).

3 The colloquial term “object” is used throughout this paper for clarity, rather than “instance,” “class,” or the more formal terms used in the DICOM standard, Information Object Definition (IOD) or Service-Object Pair Class (SOP Class).

4 Throughout the remainder of the document we will refer to the DICOM Correction Proposals (CPs) by number; all current and past CPs are archived on the DICOM Status web page (Clunie, 2016).

5 The CamelCase “keyword” form (without spaces) is used for clarity to identify DICOM data elements and attributes, rather than using the “name” or the parenthesized hexadecimal group and element tags.

6 The SR objects encoding clinical information have restricted access due to the stipulations in the consent form under which the data was collected. Before someone can access the data they need to certify that they are using the data for research purposes and that no attempt will be made to identify the individuals. These requirements were established by the TCIA team and the Washington University IRB upon reviewing the consent forms used to collect the data.

7 The API abstractions to support generation of DICOM SR documents following TID 1500 were completed after the data conversion described in this manuscript was finished. Therefore, the SR converter from the Iowa2DICOM repository referenced in the text utilizes a lower level API, which could be greatly simplified with the recent improvements to the DCMTK dcmsr module. These improvements will be implemented in the new dcmqi library.

David Clunie is the owner of PixelMed Publishing, LLC, Bangor, Pennsylvania, USA; Michael Onken is an employee of Open Connections GmbH; Jörg Riesmeier is a freelancer in Oldenburg, Germany; Steve Pieper is an employee of Isomics, Inc., Cambridge, Massachusetts, USA; and Ron Kikinis is an employee of Fraunhofer MEVIS, Bremen, Germany. The contents are solely the responsibility of the authors and do not necessarily represent the official views of the NCI/NIH.

Andriy Fedorov conceived and designed the experiments, performed the experiments, analyzed the data, contributed reagents/materials/analysis tools, prepared figures and/or tables, wrote the paper, reviewed drafts of the paper.

David Clunie, Ethan Ulrich, Christian Bauer, Andreas Wahle, Bartley Brown and Reinhard R. Beichel conceived and designed the experiments, performed the experiments, analyzed the data, contributed reagents/materials/analysis tools, wrote the paper, prepared figures and/or tables, reviewed drafts of the paper.

Michael Onken, Jörg Riesmeier and Steve Pieper conceived and designed the experiments, analyzed the data, contributed reagents/materials/analysis tools, wrote the paper, reviewed drafts of the paper.

Ron Kikinis and John Buatti conceived and designed the experiments, contributed reagents/materials/analysis tools, wrote the paper, reviewed drafts of the paper.

The following information was supplied relating to ethical approvals (i.e., approving body and any reference numbers):

University of Iowa IRB #200503706.

The following information was supplied regarding data availability:

Data: QIN-HEADNECK collection of The Cancer Imaging Archive (TCIA), access instructions: https://wiki.cancerimagingarchive.net/display/Public/QIN-HEADNECK.

Source code:

- 3D Slicer: http://github.org/Slicer/Slicer

- 3D Slicer PETDICOMExtension extension: https://github.com/QIICR/Slicer-PETDICOMExtension/tree/master/DICOMPETSUVPlugin

- 3D Slicer PETTumorSegmentation extension: https://github.com/QIICR/PETTumorSegmentation

- 3D Slicer PET-IndiC extension: https://github.com/QIICR/PET-IndiC

- DICOM TID 1500 to CSV conversion tool: https://github.com/QIICR/dsr2csv

- Iowa2DICOM conversion tools: https://github.com/QIICR/Iowa2DICOM

- dcmqi library:

https://github.com/QIICR/dcmqi

- XSLT script for converting DICOM SR TID 1500 documents to CSV list of measurements: https://github.com/QIICR/dsr2csv.

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
