# Peer review of "DICOM for quantitative imaging biomarker development: a standards based approach to sharing clinical data and structured PET/CT analysis results in head and neck cancer research"

_PeerJ, doi:10.7717/peerj.2057_

## Round 0.1 · original submission · Major Revisions

The reviewers and I all agree in that this paper describes a very relevant topic to medical image analysis research in general. It is well written, but, due to its length, is not an easy digest. We suggest to shorten or move parts of the methods body to an appendix to allow the reader to swiftly get the main message. Both reviewers address very relevant further comments that are well worth addressing.

Reviewer 1 ·

Basic reporting

Paper is written in a perfect english...

Experimental design

A relevant question is raised and analyzed. There is a need for such DICOM objects.
This is technical research.

Validity of the findings

Methods section is too long, looking more like a course.
Good point authors officially included their work in a preexisting open source work.
Results has the same format, showing at multiple times the point of view of the author ; some things that could stay in a discussion.
Discussion is fine.

Additional comments

Overall, this manuscript is very well written, and pedagogic. That's important for technical developments and helps comprenhension of these complex structures.

However, it is not conventionnally written, with a very long methods section looking more like a book chapter than a journal paper. For instance, it is the first time I see an example in a M&M section, and 900 lines...

So please remove all personal impressions from M&M and Results section ("a QI should not be like this, but like that..."), and refer to another reference (books, Internet material) for all that is already published in the standard. You could shorten the paper by third or half and make it even easier to read (that was even tough for me, a DICOM geek...).

Good job !

Reviewer 2 ·

Basic reporting

The paper addresses an import issue of research in quantitative image analysis: management and exchange of results obtained by research software tools (annotations, segmentations, quantifications, etc.) with clinical systems to allow consistent use and re-use of the data, e.g., for “validation and comparison of new approaches, secondary analysis and discovery of data, and comparison of results across sites” (lines 85/86).
The paper provide a convincing introduction and related background to its main proposition that DICOM is an appropriate choice for data exchange not only of imaging data, but also clinical information and analysis results. Relevant literature is extensively referenced.
The formal structure largely follows the PeerJ standard, with the slight deviation that the “Materials & Methods” section is entitled solely “Methods” although it also contains materials, the addition of an “implementation” section between “Methods” and “Results” (which is reasonable for a paper focusing on software methods), a “Discussion and Future Work” section and no “Conclusion”.
The “Methods” section is quite extensive (19 pages) and arguably includes not only methods, but also discussion of design decisions and limitations of the current standard. The authors themselves state that “the length of this descriptive document may be intimidating” (line 1236/1237), which especially applies to the “Methods” section. DICOM experts are probably able to easily follow the description, but the intended audience of the paper seems to be the medical image analysis community, in which the “DICOM standard is widely and fairly regarded […] as being non-trivial in complexity, while its documentation is immense and difficult to navigate” (line 179-181). If non-DICOM experts are indeed the intended audience, I suggest to more concisely presenting the finally chosen methods on not more than approx. 8-10 pages (for the “Methods” section) and moving the discussion of alternative designs, motivation of choices, and details on correction proposals to the discussion section. As an example, I found the lines 782-822 and 863-935 very enlightening, whereas I had difficulties to understand the lines 823-836. Maybe it would also be helpful to provide an example of the term “template” in the context of the DICOM standard (e.g., “SR template”), a term which is currently supposed to be known to the reader.
Some specific issues regarding basic reporting:
Lines 221-226 seem to better fit to the “Image processing” section.
Lines 267-281, 291-295: Arguably, this is not necessary for understanding the proposed DICOM-based methods.
Lines 307-315: since this does not describe a proposed method, maybe move to discussion.
Line 375: It probably has to be “DICOM PET/CT” instead of “DICOM PT/CT”.
Line 536: please provide a reference to the “PixelMed private coding scheme”
Line 640 mentions segmentations “for each such time point” whereas line 223 mentions that only for a single dataset two time points (pre-treatment and first post-treatment) have been segmented. Is this a contradiction?
Lines 692-694 mentions 4 concepts (AnatomicRegion, AnatomicRegionModifier sequence, SegmentedPropertyType, SegmentedPropertyCategory sequences) without definition, which makes it hard to understand for a non-DICOM expert. I suggest starting with examples (e.g., line 713-717) and explain the used concepts by means of the example.
Line 953: I suggest including this in the example above (lines 870-935).
Line 1102-1104: Please clarify, why it is interesting that a “private QIICR-specific template [...] actually performed a more thorough automated check”.

Experimental design

The paper is a methods/tools paper and therefore does not have a classic “experimental design”. As a methods/tools paper it is roughly similar to the paper “scikit-image: image processing in Python” (PeerJ. 2014 Jun 19;2:e453. doi: 10.7717/peerj.453), which was considered in-scope of PeerJ. Therefore, the reviewed paper can also be considered in-scope of PeerJ.

Validity of the findings

Raw data and source code is publically available. Unfortunately, the Iowa2DICOM repository (https://github.com/QIICR/Iowa2DICOM) seems to be of limited usefulness, because it applies to the conversion of a specific (presumably not publically available and not described) database, contains hard-coded directories, and is not maintained anymore according to the README on the github page (accessed Mar 5, 2016).
The paper would greatly benefit from a dedicated small set of example data (not necessarily patient data, could be phantom or artificial data) and scripts that convert this data to DICOM according to the proposed methods. This would allow reproducing the findings that the proposed methods enable a DICOM-conform representation of the data. A very good example along this line of thinking is the above mentioned paper “scikit-image: image processing in Python” (PeerJ. 2014 Jun 19;2:e453. doi: 10.7717/peerj.453), which enables the user to immediately reproduce the described methods.
A conclusion should be provided to conform to the PeerJ standard.

---

## Round 0.2 · accepted · Accept

I enjoyed reading this revision. It made the paper very readable and hopefully ensures that a broad audience gets to know about its very relevant content. A minor issue I noticed is the mentioning of 59 cases in the methods section while reporting on 60 cases.

Although I support the publication of research data, I fear the adoption of the proposed methods may be difficult. There is no win for the 'data producer' at the moment. The best the paper say now is that it hopes to be a minimal effort. It would have been nice to come up with argumentation and/or simple additional tools that that also produce the spreadsheets for the statistical analyses, rather than showing you can dump the DICOM objects. Another argument. The storage of intermediate results can be seen as additional evidence for the credibility of study results. Maybe one or two lines including such argumentation can be added to the discussion to better promote this work.

Reviewer 2 ·

Basic reporting

The paper has been carefully revised, substantially shortened, and, in my opinion, significantly improved. It is now much easier to follow. All issues that I had with the original version have been well addressed.

Experimental design

No further comments compared to the original version.

Validity of the findings

The issues raised in this context have also been fully addressed: The value of the Iowa2DICOM repository is now reasonably explained and discussed. Additionally, a new, more generally applicable tool (dcmqi, work-in-progress) is now mentioned, which – along with the other open-source tools described in the paper – will hopefully be a good starting point for readers to create their own DICOM-compliant QI data representations.